# Autologous Human Mesenchymal Stem Cell-Based Therapy in Infertility: New Strategies and Future Perspectives

**DOI:** 10.3390/biology12010108

**Published:** 2023-01-10

**Authors:** Zahirrah Begam Mohamed Rasheed, Fazlina Nordin, Wan Safwani Wan Kamarul Zaman, Yuen-Fen Tan, Nor Haslinda Abd Aziz

**Affiliations:** 1UKM Medical Molecular Biology Institute (UMBI), Jalan Yaacob Latiff, Bandar Tun Razak, Cheras, Kuala Lumpur 56000, Malaysia; 2Centre for Tissue Engineering and Regenerative Medicine (CTERM), Faculty of Medicine, Universiti Kebangsaan Malaysia, Jalan Yaacob Latiff, Bandar Tun Razak, Cheras, Kuala Lumpur 56000, Malaysia; 3Department of Biomedical Engineering, Faculty of Engineering, Universiti Malaya, Kuala Lumpur 50603, Malaysia; 4PPUKM-MAKNA Cancer Center, Universiti Kebangsaan Malaysia Medical Centre, Jalan Yaacob Latif, WPKL, Kuala Lumpur 56000, Malaysia; 5Faculty of Medicine and Health Sciences, Universiti Tunku Abdul Rahman, Sungai Long Campus, Bandar Sungai Long, Kajang 43000, Malaysia; 6Department of Obstetrics and Gynaecology, Faculty of Medicine, Universiti Kebangsaan Malaysia, Kuala Lumpur 56000, Malaysia; 7Research Laboratory of UKM Specialist Children’s Hospital, UKM Specialist Children’s Hospital, Universiti Kebangsaan Malaysia, Kuala Lumpur 56000, Malaysia

**Keywords:** infertility, mesenchymal stem cells, assisted reproductive technology

## Abstract

**Simple Summary:**

Infertility is a global issue and the currently available treatments for infertility are known to pose some risks and are lacking in solving infertility problems that are closely related to genetic disorders. In this review, we summarized the use of autologous mesenchymal stem cells (MSCs) as the cell-based treatment for infertility and the future treatments of infertility using various methods.

**Abstract:**

Infertility could be associated with a few factors including problems with physical and mental health, hormonal imbalances, lifestyles, and genetic factors. Given that there is a concern about the rise of infertility globally, increased focus has been given to its treatment for the last several decades. Traditional assisted reproductive technology (ART) has been the prime option for many years in solving various cases of infertility; however, it contains significant risks and does not solve the fundamental problem of infertility such as genetic disorders. Attention toward the utilization of MSCs has been widely regarded as a promising option in the development of stem-cell-based infertility treatments. This narrative review briefly presents the challenges in the current ART treatment of infertility and the various potential applications of autologous MSCs in the treatment of these reproductive diseases.

## 1. Introduction

In clinical practice, infertility is defined as a disease of the male or female reproductive system in failure of conceiving within 12 months of regular, unprotected sexual intercourse in women less than 35 years old or within 6 months in women 35 years and older [1]. According to the latest data from World Health Organization, this condition affects 48 million reproductive-age couples worldwide and is categorized into primary and secondary infertility. Primary infertility is when a person has never achieved a successful pregnancy while secondary infertility is when a person is having difficulty getting pregnant after a prior successful pregnancy [2]. The general perception of infertility is predominantly caused by women; however, the causes are equally distributed among men and women. Men and women each contributed 40% as a sole or a contributing cause of infertility, and in the remaining 20%, it is the male and the female partner or unidentifiable cause termed unexplained infertility [3].

An increasing number of infertile couples are turning to ART as the most effective way in treating infertility in humans. Nonetheless, the presence of a few limitations within this method has diverted scientists’ attention toward stem cells, particularly germ-like stem cells in the treatment of infertility [4,5,6,7,8]. Research on MSCs is intriguing compared to ESCs that involve the destruction of a human embryo which sparked some serious ethical issues debate [9,10]. Therefore, spindle-shaped cells of MSCs with the ability to differentiate into multiple germ layers coupled with the unique characteristics of MSCs have been suggested as an ideal candidate for regenerative medicine [11]. Different reports have presented solid evidence on the role of MSCs in the recovery of fertility and the future outlook of infertility treatment [8,12,13]. However, due to the immunogenicity and the risk of anti-donor immune response with allogeneic MSCs application, this present review specifically focuses on the advantage of only using autologous MSCs as a safer treatment of infertility, which was not discussed by previous reports. 

## 2. Causes and Mechanisms Leading to Infertility

Couples that fail to conceive within a year will usually undergo initial evaluation for the standard clinical diagnosis that includes semen analysis, assessment of ovulation, and tubal patency test [14]. In 20% of the cases, this standard fertility evaluation fails to identify any abnormalities, termed unexplained infertility [15]. In known causes among infertile couples, there are various environmental and internal identifiable factors that play a role including ovulatory dysfunction, tubal diseases, male infertility factors, lifestyle, and environmental factors such as smoking and obesity that can adversely affect fertility (Table 1) [16,17,18,19]. Most of these factors impacted several biological systems such as the neuroendocrine systems, the reproductive organs, and the immature and growing follicles or sperm. The most known common factors in infertility include infection [20,21,22], congenital defects [23,24], and gonadotoxins [25,26,27,28]. 

## 3. Limitations in The Current Conception Treatment 

The treatments received by couples depend on the cause of the problems. Couples with presenting fertility issues will either receive medication, surgical procedures, and/or in combination with assisted conception such as intrauterine insemination (IUI), in vitro fertilization (IVF), or intracytoplasmic sperm injection (ICSI) [14]. Medications that are commonly used to invoke ovulation include clomiphene, letrozole, tamoxifen, metformin, and gonadotropins [29]. On the other hand, women usually undergo surgical procedures on fallopian tubes with a presentation of fibroid, polycystic ovarian syndromes (PCOS), or endometriosis issues. In men with varicocele [30], a corrective surgical procedure will be performed and sperm retrieval will be done on the block epididymal [31]. On the other hand, the gold standard treatment for cancer and non-cancerous patients that need to undergo chemotherapy treatments involves cryopreservation of the gonadal tissues [32]. Many known complications are associated with ART despite it being envisioned as the solution for infertile couples. 

### 3.1. Multifetal Gestations and The Effects 

ART is a known risk for conceiving higher multifetal gestations due to the transferring of more than one embryo. From a total of 1.8% of infants born using ART in the United States, ART procedures resulted in 16.4% of multiple gestations with 16.2% of twins and 19.4% of triplets or higher order pregnancies [33]. Some of these ART procedures are accompanied by clomiphene, letrozole, and gonadotrophin as oral ovarian induction agents, which show to possess a risk of multiple gestations based on a review [34]. Multifetal gestation also leads to various maternal complications such as birth defects [35] and stillbirths [36], which imposed serious effects on the fetus. Besides that, maternal complications are also present in singleton births conceived via ART. A comparison of singleton births between naturally conceived and ART revealed a significantly low birth weight and preterm birth [37] with 1.1% extreme preterm birth (20–27 gestational weeks) and 2.2% very preterm birth (28–32 gestational weeks) in ART conceived babies [38]. The double risk of cardiovascular, genitourinary, and musculoskeletal systems on the fetus after a cycle of ICSI has been reviewed [39]. This is a worrying fact since it consists of 70% of ART treatments worldwide [40]. 

### 3.2. Ectopic Pregnancy 

Ectopic pregnancy following infertility treatments is one of the risk factors of ART [41], which has been also associated with significant morbidity and mortality [42]. It was reported that the incidence is <5% in ART pregnancies [43]. Tubal factor infertility is suggested to contribute to the highest percentage of ectopic pregnancy up to 11% among infertile patients [44,45] and the highest rate is after tubal reconstructive surgery [46]. Hormonal changes that happen after infertility treatments could also affect the expression of signaling molecules that are needed for the interaction between the embryo, Fallopian tube, and endometrium [47,48]. Evidence also exists on the increased risk of ectopic pregnancy due to the immunological changes after embryo transfer [49]. 

### 3.3. Ovarian Hyperstimulation Syndrome (OHSS)

The development of OHSS during ART is an iatrogenic side effect of using high-dose ovary stimulatory agents, mainly gonadotrophin [50], and can be fatal. This often resulted in multi-follicular growth characterized by abdominal tenderness, nausea, vomiting, renal failure, and swelling due to increased ovarian volume. After decades, an alternative stimulation to triggering final oocyte maturation has eliminated the risk of severe OHS, nonetheless, it reduces the probability of achieving pregnancy [51].

### 3.4. Birth Defects

Major congenital malformations, aneuploidy, and mosaicism are examples of worrying issues in IVF and ICSI [52,53,54]. The high prevalence of this occurrence could be due to the transferring of poor embryos during the procedure [55] and the most well-documented evidence is the twinning effects or higher-order pregnancies [56]. Much debate also revolves around the possible factor of ovulation induction [56,57] and the micromanipulation of the techniques themselves [58], which resulted in injuries to the internal structure of oocytes leading to deleterious consequences such as aneuploidy and chromosomal abnormalities.

### 3.5. Cryopreservation of Testes and Ovary

Cryopreservation of testicular or ovarian tissue is one method of fertility preservation in childhood cancer patients [59,60]. Thus far, all ovary transplanted studies reported freezing and ischemic injury limitations. Transplantation resulted in far greater ovarian follicle damage compared to cryopreservation due to ischemic and oxygen deficit [61]. Using a sheep model, the survival of the primordial follicles and the formation of secondary follicles on different sites have been successfully established [62]. However, according to the study, a longer experimental study evaluating hormone recovery, follicular maturation, and successful pregnancy are critical for clinical reference. 

## 4. MSCs and Their Mechanism in The Treatment of Infertility 

With the shortcomings of the ART treatments, the heightened pathological immune response following infection and inflammation leading to deleterious effects on fertility, and gonadal disorders that are seen to impede normal pregnancy necessitates a good balance between immune activation and immune suppression, and tissue replenishment. A long search for an alternative treatment in correcting limitations in ART or conditions that are untreatable through ART has set stem cells as an appealing new treatment with the possession of several biological characteristics that qualify them to be used for cellular therapy and new hope in infertility treatments. 

Stem cells have the ability to renew themselves for long periods without significant changes in their general properties. Under certain physiological or environmental conditions, these cells are able to differentiate into various specialized cells. Despite the research excitement on embryonic stem cells (ESCs) and induced pluripotent stem cells (iPSCs), they possess ethical concerns, immune rejection after transplantation, and teratoma formation issues. This exposes MSCs and tissue-specific stem cells as an area of great interest [63] since they can be expanded and manipulated ex vivo. MSCs are a subset of non-hematopoietic adult stem cells that originate from the mesoderm and possess self-renewal abilities and multilineage differentiation. MSCs can differentiate into mesoderm lineages [64,65,66] and are harvested from multiple adult tissues including the following: skeletal muscle, cervical tissue [67], menstrual blood [68], bone marrow, adipose tissue [69], umbilical cord [70], umbilical cord blood [71], amnion [72], placenta [73], and fetal tissues such as blood, liver, and bone marrow [74]. While tissue-specific stem cells are derived from reproductive organs known as germline stem cells such as from the testis [75] and ovary [76] for the continuous production of sperm and oocytes. The precise mechanisms of MSCs are still not fully understood. However, in relation to infertility possible actions, the four most prominent are their biological characteristics of differentiation, secretory capacity, mitochondrial transfer, and immunomodulatory and anti-inflammatory capacity (Figure 1): 

***Differentiative capacity*.** MSCs are capable of differentiating, albeit limited to the mesoderm layer, into various cell types such as epithelial, stromal, and endothelial cells [77]. This highlighted the potential of MSCs for advanced tissue repair treatments for infertile couples with damaged endometrial [78], restoring endometrial function [77], or ovarian tissues [79]. Although according to some studies that MSCs improve and help ovarian function recovery, the number of differentiated and functionally integrated MSCs is too small to observe improvements in ovarian function. However, it remains unclear on the mechanism of MSCs differentiation into target cells such as oocytes or supporting cells after migrating to the injured tissues to improve and correct ovarian dysfunction [80]. A new hypothesis also emerges by indicating that rather than assuming the engraftment and differentiation of MSC in diseased tissues or organs, an alternative mode of rescue and repair could enhance the cell viability and proliferation responses of MSC, which is discussed in the point below.

***Secretory capacity*.** At present, this new hypothesis is widely accepted among researchers in which MSCs’ effect on reproductive treatment thus far is linked to various bioactive secretome factors such as insulin-like growth factor (IGF), vascular endothelial growth factor (VEGF), cytokines, and other growth factors [81,82]. Nevertheless, the importance of the paracrine effects of MSCs and their secretomes in restoring the cellular composition of tissues via regulating the immune response, stimulating angiogenesis, and maintaining the viability of the microenvironment [83] is highlighted by several groups. Some evidence includes the paracrine activity mechanisms driving the improved ovarian function [84,85] and endometrial reserve [86,87] than its stimulatory effects on cell growth and differentiation. For example, the paracrine activity of VEGF, as a strong proangiogenic factor in ovary vascularization [88], secreted by MSCs has been shown to enhance ovarian function by restoring its structure [89]. Apart from the secretomes, some miRNAs and exosomes carried by the MSCs have been found to be useful biomarkers in targeting infertility issues. miRNA-644-5p and miRNA-144-5p carried by bone marrow-derived mesenchymal stem cells (BMSCs)-derived exosomes have been shown to promote the recovery of ovarian function in chemotherapy-induced POI in a rat model [90,91]. As well in men, MSC-derived exosomes are able to induce spermatogenesis in the testes of infertile azoospermic mice models [92]. These findings underpin the role of the gene expression regulated by miRNA in MSC-based therapy outcomes and its importance in regulating stem cells self-renewal and differentiation by repressing selected mRNA translation [93].

***Mitochondrial transfer*.** In reproduction, mitochondria are transmitted to the offspring exclusively by the oocytes from the mother. This organelle is important for optimal oocyte quality, proper fertilization, and embryo development [94]. Therefore, the mitochondrial transfer (MT) technique is seen as an utmost strategy for improving oocyte quality in women with a history of poor oocyte quality, advanced maternal age women, and patients with previous IVF failures which shared defects at the oocyte level. MT can be accomplished heterologously (using a donor oocyte) or autologously from ovarian stem cells or granulosa cells [95]. However, heterologous MT introduces a third source of DNA [96] while the existence of ovarian stem cells remains questionable [97] and granulosa cells undergo an aging process along with the oocyte [98]. In an animal model of aged mice, autologous adipose-derived stem cells MT rescues the oocyte quality with a significant mitochondrial oocyte difference between the young and aged mice [98], while another study found no advantage of MT [99]. This difference in the observation could be explained by the different genetic backgrounds of the mice. There is also a growing body of evidence on the ability of Sertoli cells, which is now believed to possess some MSCs characteristic [100], to exhibit the properties of MT [101]. Acknowledging the lack of studies on MSCs MT in infertility, a review study highlighted few studies on the importance of MT from MSCs in restoring normal physiological function and disease recovery [102]. Therefore, with extensive further research using in vivo models and human studies, MT could provide a valid treatment strategy for low fertility or infertility in women and men. While MSCs can give some effect of cell-to-cell interaction via MT, their secreted properties of MSCs via exosome can help to modulate the immune system. 

***Immunomodulatory and anti-inflammatory capacity*.** MSCs, via their exosomes, have been shown to possess broad immunoregulatory abilities by influencing both adaptive and innate immune responses. Findings showed that MSCs can inhibit T cell proliferation and conversion to regulatory T cells (Tregs) through the reprogramming of M1 macrophage cells to M2 phenotype, leading to tissue repair and healing [103,104]. PCOS, as one of the most common endocrine–reproductive–metabolic disorders in women [105], exhibited higher Th1 inflammatory responses [106,107]. Therefore, the capability of MSCs to suppress Th1 may influence the internal PCOS inflammatory environment hence restoring ovarian function via the cessation of the autoimmune reaction and the regression of the endocrine disease [108]. Bacterial infection-induced pelvic inflammatory animal models showed a promising immunomodulatory role of MSCs in partially restoring fertility, in which the MSCs repair the tubal epithelium structure subjected to chronic inflammation, decreasing the inflammatory factors, and restoring the oviductal glycoprotein secretion level [109,110]. In the same study, inducing *E. coli* in rabbits showed decreased inflammatory factors and increased oviductal glycoprotein expression, reflecting improved sperm fertilization capacity [109]. Improved pregnancy rate, ovary morphology, and apoptotic oocytes were also observed in inflammation-induced ovary mice [111]. Antisperm antibody (ASA) that is naturally present in the fertile population or due to traumatic testis rupture [112] resulted in autoimmunization against spermatozoa in the form of a humoral immune response. An in vivo mice model study revealed that MSCs are able to suppress the production of ASA by modulating the humoral immune response [113] thus increasing sperm concentration and motility. Considering that COVID-19 virus infection affected male fertility by reducing sperm count and motility [114], studies on MSCs could be advantageous in COVID-19-induced infection and inflammation [115] and semen quality. 

## 5. Potential Usage of MSCs in Infertility 

ART is undoubtedly a well-recognized method for achieving pregnancy for infertile couples. Despite that, it presents challenges to public health as evidenced by the high rate of multifetal gestation, preterm delivery, and low birth weight of infants [116]. Exhaustion of traditional treatments and genetic defects that lead to gamete deficiency have led to mounting preclinical studies on animals and clinical studies on humans using stem cells one of which is using MSCs. This MSCs therapy-based technology could generate auxiliary factors in improving the functional roles of various cell types in the reproductive systems. Manipulation of BMSCs, umbilical cord stem cells, amniotic fluid mesenchymal stem cells, menstrual stem cells (MenSCs), adipose-derived stem cells (ADSCs), and endometrial MSCs with the options using autologous or allogeneic treatments proved the effectiveness of MSCs in infertility. The success of these interventions in pre-clinical and clinical studies has brought huge hope in improving female and male reproductive health [12]. Nonetheless, autologous treatment is favorable among researchers for several reasons. Following MSCs therapy, studies reported the potential of immunogenicity after allogeneic applications [117,118,119]. In one of the studies, in the effort of preserving infarcted heart function using allogeneic MSCs, a biphasic immune response was seen after 5 months of implantation suggesting the MSCs transition from immunoprivilege to an immunogenic state [120]. Due to the potential risk of anti-donor immune responses, several strategies were suggested in a systematic review including the use of immunosuppressive drugs in combination with MSCs therapy [121]. To date, the risks and limitations of both autologous and allogeneic therapeutic applications are highly debated such as in terms of the potential impact of donor-donor heterogeneity. Since there is a lack of review studies on the autologous applications of MSCs in infertility treatment, in this section, we focus on the available autologous MSCs treatments for infertility issues, completed or ongoing, in both animal models and human clinical trials.

### 5.1. Preclinical Studies

Current in vitro studies involve using MSCs alone or in combination with other drugs or stimulants for the potential application of ovarian dysfunction and endometrial disorders in females and spermatogenesis in males. ADSCs are a type of MSCs that can be easily isolated and collected in a minimally invasive procedure and an abundant in quantities. It is also safe to be transplanted autologously into a host [122]. Apart from the whole cells, mitochondria from ADSCs have also been used to correct aging as a predisposing factor to fertility health using an animal study [98]. The microinjection of ADSCs mitochondria promotes oocyte quality, embryonic development, and fertility in elderly mice thus promising an exciting strategy for elderly women. 

Since the concept of pluripotent stem cells could differentiate into functional gametes, few germline stem cells have been associated with the recruitment and stimulation or conversion into functional gametes [123]. These extragonadal sources of gametes could aid in the hampered spermatogenesis by testicular damage or oogenesis in producing healthy eggs. An earlier study on autologous transplantation of spermatogonial stem cells (SSC) done on monkeys showed positive results on the structure of the testes [124], yet no evidence on sperm quality. A later study using macaque supported the finding with the same increase in the size of the testes and evidence of the successful production of healthy sperm [125]. Table 2 shows a list of autologous MSCs stem cell transplantation for various infertility issues using animal models.

### 5.2. Clinical Studies

Translating the fundamental approach of preclinical studies to human studies is challenging. Most clinical trials have yet to acquire full regulatory approval thus hindering the promotion of stem cells into clinical practice. Despite that, there is encouraging news that could pave a way for the use of MSCs in infertility treatments in the future. One successful intervention was the birth of a baby after transplanting autologous BMSCs into an ovary of a POI woman [128]. However, there is a lack of data on serum hormone levels, MSC preparation details, and imaging data. A pilot result from a clinical trial fulfilled these void data and presented a report showing a 50% increase in ovarian volume compared to the atrophic contralateral ovary and a 150% increase in estrogen level. This trial showed enhanced fertility health with a resumption of menses and diminished menopausal symptoms [129]. Successful transplantation of BMSCs using various methods in intrauterine adhesion (IUA) or known as Asherman’s syndrome showed positive outcomes for clinical practice [130,131,132,133,134]. Subendometrial transplantation of BMSCs resulted in menstruation restoration in 5 of 6 IUA cases [132]. In addition, BMSCs also could be a promising therapy for advanced reproductive-age women with ovarian reserve issues [135]. Loss of the anti-Mullerian hormone and antral follicle count that is associated with aging is restored with BMSCs potentially via the homing ability of stem cells. Together with other multiple paracrine factors, these cells differentiate into a variety of cells to facilitate ovarian recovery [136]. In azoospermic condition, a promising result was obtained in a pilot clinical study [137] with various clinical trials being performed or are underway on the injection of BMSCs to the rete testis to assess the hormonal level, testicular size, and sexual potency (Table 3). 

ADSCs clinical trials have been performed in POI [138], IUA [139,140], azoospermia, and post-prostate cancer treatment for erectile dysfunction [141]. Subendometrial injection in women with thin endometrium lining is associated with a total of 13 pregnancies and 9 live births [140], while subendometrial transplantation increases endometrium lining to 7 mm in 3 out of 6 cases [139]. This small cohort of patients granted replication and expansion before drawing any conclusion since these trials have not resulted in complete positive outcomes for most of the participants. In POI treatments, each of the patients in the trial experienced variable ovarian volume, anti-Mullerian hormone, and antral follicular count after transplantation. Although menstruation was resumed in two patients, the other patients showed no improvement, while all patients demonstrated inconsistent FSH levels [138]. Meanwhile, only 47% of men showed recovered erectile dysfunction and were able to accomplish sexual intercourse after stem cell transplantation [141]. 

Another promising trial is the transplantation of autologous MenSCs for severe IUA syndrome patients. Two independent research studies demonstrated improved endometrial thickness in women with thin endometrium. One study reported increased endometrial thickness to 7 mm and an improved pregnancy rate between 47% [142] to 50% [143]. These data open new hope for infertility female patients caused by IUA with the ease of sample collection and less invasive method compared to other sources of MSCs. As reported in a reviewed study, almost 130 cryopreservation have been conducted worldwide. However, frozen ovarian tissue cryopreservation transplantation showed evidence of relapse or re-introduction of cancer cells [144] and needs further improvement in pregnancy rates [145].

Clinical use of endometrium MSCs in a case presentation of a woman with multiple failed ART cycles successfully increases the endometrium receptivity of the woman. Ultrasound examination prior to embryo transfer showed that the endometrium thickness improves 2.15-fold after endometrial MSCs transplant [146] and the patient achieved a live successful pregnancy. Despite the increase in the endometrium thickness not within the optimal range, a cocktail of growth factors, cytokines, and hormones may assist in the endometrial receptivity for pregnancy [147]. 

**Table 3 biology-12-00108-t003:** Clinical trials of autologous stem cells treatment for infertility issues.

Cell Source	Disease	Mode of Treatment	Outcome	Reference/ NCT ID
BMSCs	IUA (Asherman’s syndrome)	Transplantation in the endometrial cavity	Restoration of the menstrual cycle	[132]
			Increased endometrium thickness and good vascularity	[130]
	IUA (Asherman’s syndrome)	Intra-arterial to the uterus	Increase the volume and duration of menses Increase the thickness of endometrium and angiogenesis process Decrease IUA score Spontaneous conception	[131]
	IUA (Asherman’s syndrome)	BMSCs-loaded collagen scaffold	Restore endometrial regeneration Increased successful pregnancy and live births	[133]
	IUA (Asherman’s syndrome)	Transplantation to the uterine cavity	Recovered endometrium	[148]
	POI	Laparoscopic instillation into ovaries	Improved AMH level Successful pregnancy with live birth	[149]
		Injection into the ovary	Successful pregnancy in 1 patient out of 10 Regained menstruation	[128]
			Resumed ovarian estrogen production Resumed menses	[129]
			Increased endometrial thickness Normalization of FSH level Pregnancy occurs within 12 months of follow-up Recovered folliculogenesis	NCT03069209
			Elevation in serum AMH and estrogen level Decline in serum FSH level Disappearance of menopausal symptoms	NCT02043743
			Elevation in serum AMH and estrogen level Decline in serum FSH level	NCT02062931
		Injection via peripheral vein	No results posted (unknown status)	NCT02779374
	Azoospermia	Injection into rete testis	Increased testicular size Elevation of testosterone level Reduction of FSH level	[137]
	Azoospermia	Intra-testicular transplantation	No results posted (recruiting)	NCT02641769
	Azoospermia (Klinefelter Syndrome)	Injection into testicular tubules and artery	No results posted (recruiting)	NCT02414295
	Non-obstructive azoospermia	Injection into testis	No results posted (recruiting)	NCT02041910
	Non-obstructive azoospermia	Injection into testis	No results posted (recruiting)	NCT02008799
	Ovarian reserve	Intra-ovarian artery injection	Increased antral follicular count Increase in anti-mullerian hormone Improved ovarian function	[135]
ADSCs	Thin endometrium syndrome	Subendometrial injection	Increased endometrium thickness Increased successful pregnancy and live birth	[140]
	IUA (Asherman’s syndrome)	Transcervical instillation	Resume menstruation in amenorrhea Higher menstruation amount in oligomenorrhea	[139]
	Azoospermia and oligozoospermia	Injection into testis	No results posted (Enrolling by invitation)	NCT03762967
	POI	Intra ovarian transplantation	Resumption of menses Decreased FSH serum levels Variable ovarian volume, anti-Mullerian hormone, and antral follicular count	[138]
	POI and Ovarian Ageing	Injection into ovary	Increased number of follicles Increased number of blastocysts Improved endometrial thickness	[150]
	Post-cancer surgical removal of erectile dysfunction	Single intracavernous injection	Improved erectile function	[141]
MenSCs	IUA (Asherman’s syndrome)	Transplanted into uterus	Regenerating the endometrium, prolonging menstrual duration, and increasing the rate of pregnancy	[142]
		Transplantation	Increased endometrial thickness Improved pregnancy	[143]
Endometrial-MSCs	Thin endometrium syndrome	Submucosal injection	Increased endometrium thickness Increased successful pregnancy and live birth	[146]

## 6. New Strategies and Future Perspectives

There are hundreds of registered clinical trials trying to explore multipotent MSCs in imaginable infertility treatments for clinical application. However, these clinical-stage MSCs therapies are unable to meet the primary efficacy endpoints as their administration in humans is not as robust as demonstrated in preclinical studies. Meanwhile, the translation of cell-based therapy is impaired by the biological differences between normal and stem cells from the same tissues (heterogeneity population) and it is not a straightforward application. Therefore, leveraging other possible perspectives are needed to achieve more potent and versatile autologous therapies as proposed in Figure 2. 

### 6.1. Cell-Free Therapy

Although autologous stem cell treatments have been widely used in reproductive medicine due to their promising properties, extensive clinical application is impeded by its safety, high cost, low quality, and manufacturing. In recent years, mounting evidence has emerged linking the secretion of extracellular vesicles (EVs) such as exosomes from MSCs as the main driver of the mechanism of action. Although the isolation methods involve differential centrifugation [151], extracting these exosomes from low-immunogenic MSCs could solve challenges associated with autologous or allogeneic stem cell treatments. As such, this has attracted the attention of researchers in making use of these cell-derived EVs with a concept known as cell-free therapy. 

#### 6.1.1. MSCs-Derived Exosomes/miRNA

In understanding the potential use of the miRNA-derived exosomes, miRNA-21 was overexpressed in the MSCs. Upregulation of the miRNA-21 increases the estrogen, decreases the FSH level, and decreases granulosa cells apoptosis by downregulating the programmed cell death protein 4 (PDCD4) and phosphatase and tensin homolog (PTEN) [152]. The full potential of the exosomes derived from MSCs is reflected in multiple studies. Using the POI animal model, the ovarian function was restored by the downregulation of PDCD4 and PTEN and regulation of reproductive hormone levels via miRNA-155-5p [91] while miRNA-644-5p acted by targeting the p53 [90]. In a model of IUA, miRNA-29a [153] and miRNA-340 [154] have been shown to inhibit fibrosis during an endometrial repair. Although the exact mechanism is unknown, it is speculated that the exosomes reverse the epithelial-mesenchymal transition (EMT) and promote endometrial repair via TGF-β1/Smad signaling pathway [155] and this was further proved in cutaneous wound healing [156]. In addition to possessing a repair mechanism, the exosomes exhibit a protective effect against sperm genomic integrity injuries (such as cell membrane injury and DNA damage) and ROS [157], which raises the possibility of exosome therapy for asthenozoospermia.

#### 6.1.2. ADSCs-Derived Exosomes/miRNA

Few studies also explored the outcomes of exosomes derived from ADSCs in a few infertility issues. One study demonstrated that ADSC-exosome promoted endometrial regeneration and receptivity thus restoring fertility [158]. Transplantation of the exosomes in the tissue grafts of the POI model, on the other hand, showed decreased apoptosis by downregulating the Fas ligand and increased the SMAD5 expression [159]. Both notions hold great promise in addressing implantation and pregnancy facilitation in infertile patients. Using a PCOS model, miRNA-323-3p extracted from modified ADSCs provided new insight into developing new strategies for PCOS patients since the miRNA promoted cell proliferation and inhibit cumulus cell apoptosis [134]. The ADSCs-derived exosomes were also explored in male infertility whereby in a diabetic rat model, transplantation of the exosomes ameliorates erectile function [160], suggesting possible erectile correction in human infertile males. 

#### 6.1.3. Menstrual SCs-Derived Exosomes/miRNA

Results from the transplantation of MenSCs on the POI animal model were successfully replicated in the in vivo and in vitro MenSCs-derived exosomes study. Exosome exposure inhibits follicle apoptosis and promotes the proliferation of granulosa cells while in vivo models presented with promoted follicle development, restored estrous cycle and serum hormone levels, and improved live birth [161]. This exciting outcome suggests a desirable cell-free bioresource in infertility treatments. Applying exosomes to living tissues has grabbed the focus as the future therapeutic effects since it does not induce inflammation, teratomas, and degraded by enzymes. Many potentials of the exosome’s effects on tissue engineering, regenerative, and reproductive medicine [162] have been depicted, which could be the answer to filling the gap in clinical trials-bedside of infertility treatment. 

### 6.2. Very Small Embryonic-like Stem Cells (VSELs)

Over a decade, a small population of small, early-development stem cells known as VSELs was identified as pluripotent stem cells based on their primitive morphology and gene expression profiles [163,164]. Researchers proposed that these cells originate from the germ line, are deposited in the developing organs during embryogenesis, and play a crucial role as a backup population for monopotent tissue-committed stem cells. VSELs are highly quiescent when residing in the adult tissues due to the erasure of the regulatory sequences for certain imprinted genes [165] and they possess large nuclei containing euchromatin and a thin rim of cytoplasm enriched in spherical mitochondria. During stress situations or induced to proliferate, they are activated and released into circulation (as reviewed in [166]). Although these cells are evolutionarily conserved in mammals (reviewed in [167]) and demonstrated in a small number, a recent study has successfully expanded the cells ex vivo without feeder layer cells while preserving their capacity to differentiate into organ-specific cells [168]. Convincingly as well, these cells can be isolated from the testes [169] and ovarian surface epithelium of young and postmenopausal women [170], which can differentiate into oocyte-like cells in response to sperm cells and release zona pellucida [171]. The existence of these cells in azoospermic testicular biopsies of an adult male cancer survivor [172] and busulfan and cyclophosphamide ovaries-treated mouse model [173] is anticipated to open a new avenue for fertility restoration in cancer survivors. Despite the very existence of these cells being highly questionable among the experts in the field, it is hopeful that these cells can be the new hope for the collection of autologous pluripotent stem cells treatments in infertility issues such as delaying menopause and most importantly enable aged mothers to have better egg quality. 

### 6.3. Regenerative Therapy

Furthermore, the use of microfluidic chip devices and BMSCs and ADMSCs has been able to promote stem cell maintenance and differentiation into functional organ models [174]. Although this novel study of stem cells and microfluidic chip-based model was performed on a bone, this has created an interesting area to be explored and manipulated on reproductive organs to understand their physiological function and disease modeling before transplantation. This chip-based model in turn could provide a unique opportunity for infertile patients with impaired gametogenesis caused by congenital disorders in sex development or cancer survivors [175] that is lacking in the current infertility treatments. One recent study has taken a step forward in generating regenerative therapies in the treatment of type 1 diabetes using scalable GMP-grade human pancreas organoids [176]. A similar approach could be devised in infertility treatments by developing endometrium organoids from small endometrial biopsies and transplanting autologously to restore damaged epithelium hence would avoid allogeneic immune responses.

**Figure 2 biology-12-00108-f002:**
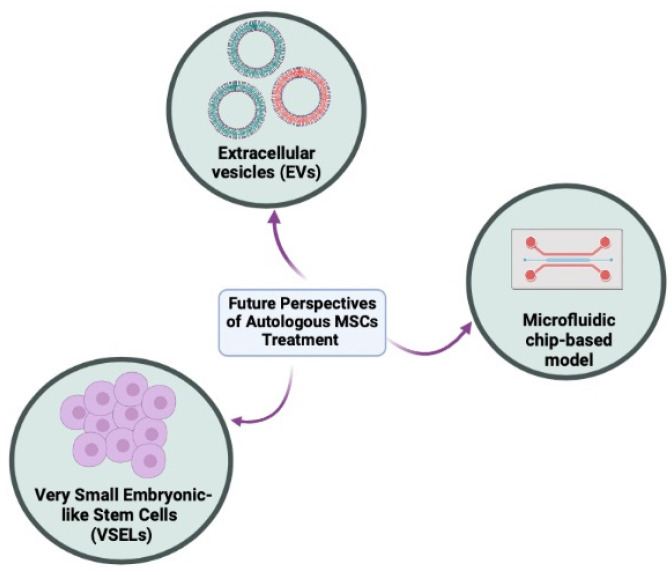
The future autologous methods in tackling infertility issues in men and women. These methods include cell-free therapy using extracellular vesicles (EVs), very small embryonic-like stem cells (VSELs), and organoids or functional organ-on-a-chip (OOAC) models (Created with Biorender.com).

## 7. Conclusion

In summary, stem cells provide an exciting opportunity in developing potential new treatments for infertility in both men and women. Both allogeneic and autologous MSCs are the key player in cell-based therapy; however, autologous stem cell treatment is regarded as safer and immunoprivileged. The combined effects of MSCs transplantation and the secretomes or exosomes predominantly play an important role in the recovery of failing reproductive tissues or organs. Considering that many MSCs are in the preclinical investigational stage, the progress in using these potential MSCs as stem cell therapy requires further long-term planning with strict evaluation and supervision to ensure accuracy, quality, and safety before implementing these approaches at the bedside. The emergence of invaluable cell sources such as VSELs, EVs, and the cutting-edge technology of organoids could offer promise in the quest to overcome human infertility.

## Figures and Tables

**Figure 1 biology-12-00108-f001:**
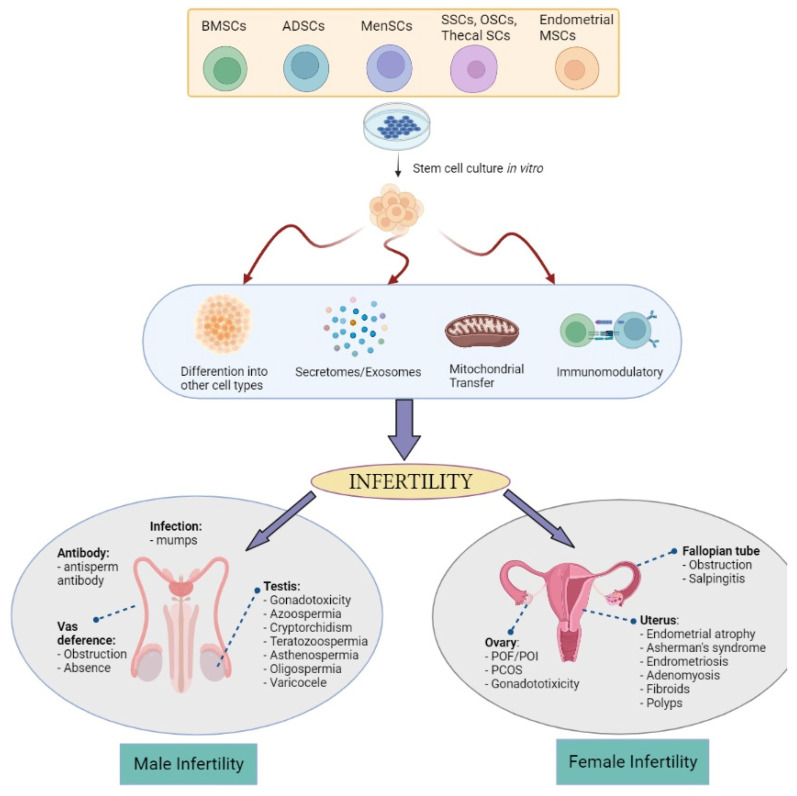
The possible mechanism of actions of bone-marrow stem cells (BMSCs), adipose tissue stem cells (ADSCs), menstrual stem cells (Men-SCs), endometrial MSCs, and germline-derived stem cells: spermatogonial stem cells (SSCs), oogonial stem cells (OSCs), and thecal stem cells in treating various male and female infertility problems (Created with Biorender.com).

**Table 1 biology-12-00108-t001:** Common causes of infertility in male and female.

Common Causes of Male Infertility	Common Causes of Female Infertility
Hypothalamic hypophyseal causes	Ovulation Disorders Causes
Pituitary insufficiency	Ageing (diminished ovarian reserve)
Hyperprolactinemia	Premature ovarian failure/insufficiency (POF/POI)
Kallmann’s syndrome (anosmia)	Endocrine disorders (such as PCOS)
Testicular Disorders	Tubal Causes
Klinefelter syndrome	Pelvic inflammatory disease
Chromosome anomalies (AZF microdeletions)	Tubal surgery
Testicular atrophy	Previous ectopic pregnancy
Varicocele (excessive heats)	Salpingectomy
Cryptorchidism	Uterine/Cervical Causes
Infections (such as mumps)	Congenital uterine anomaly
Disorders of the Seminal Tract	Fibroids (Asherman’s syndrome)
Retrograde Ejaculation	Endometriosis
Obstructive azoospermia	Poor cervical mucus quality/quantity
Immunological causes	Infection (salpingitis)
Autoimmunity to sperm	Others
	Tumours/treatment
	Obesity
	Environment

**Table 2 biology-12-00108-t002:** Animal models of autologous stem cell treatment for infertility issues.

Cell Source	Disease	Mode of Treatment	Model	Outcome	Reference
ADSCs	POI (aging mice)	Microinjection into oocytes	Mice	Reduction of aneuploidy rates in the eggs Improve the quality of mature eggs Promote embryo development Rescue fertility in aged mice	[98]
	POI (aging mice)	Microinjection into oocytes	Mice	Did not mitigate the poor fertilization and embryonic development rates of aged oocytes	[99]
	POI	Melatonin-pretreated intraovarian injection	Mouse	Recovered serum hormone levels and reproductive function Increased primordial follicle mean counts	[126]
BMSCs	Testicular damage due to chemotherapy	Injection into the testes	Rats	Homing of the stem cells at the germinal epithelium Differentiate into spermatogonia cells Did not differentiate into Sertoli cells	[127]

ADSCs—adipose tissue-derived stem cell; BMSCs—bone marrow-derived stem cell.

## Data Availability

Not applicable.

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
