# Peer review of "Autologous Human Mesenchymal Stem Cell-Based Therapy in Infertility: New Strategies and Future Perspectives"

_biology, 2023, doi:10.3390/biology12010108_

Round 1
Reviewer 1 Report
1. Please write the full name the abbreviation appears the first time.
2. Some references are too long, so it is recommended to modify them.
Author Response
Response to Reviewer 1 Comments
Point 1: Please write the full name the abbreviation appears the first time.
Thank you for taking your time in reviewing our manuscript. We have made the necessary changes in terms of the abbreviation.
Point 2: Some references are too long, so it is recommended to modify them.
Thank you for your comment. We have changed the reference style according to the guideline (by using MDPI Endnote style).
Reviewer 2 Report
The authors presented a manuscript summarizing current knowledge on the use of mesenchymal stem cells (MSCs) in female and male infertility therapies. Overall, this topic is important, however, several articles have been published recently that review the current knowledge on this topic. For example? Fazeli et al. Mesenchymal Stem Cells (MSCs) Therapy for Recovery of Fertility: a Systematic Review DOI10.1007/s12015-017-9765-x. Saeed et al Mesenchymal stem cells to treat female infertility; future perspective and challenges: A review DOI10.18502/ijrm.v20i9.12061; Saha et al. Application of Stem Cell Therapy for Infertility DOI10.3390/cells10071613; Jahanabani et al. Stem cells technology as a platform for generating reproductive system organoids and treatment of infertility-related diseases DOI10.1002/cbin.11747. (These articles are not mentioned in the presented manuscript!) Therefore, it is very difficult to find a specific field that has not been covered by recent reviews and in my opinion, the authors failed in this respect.
The introduction of diseases leading to infertility is too long and is not associated with the topic – using of MSCs. The necessity of stem cell application in the treatment of both female and male infertility is supported by the articles focusing only on male infertility (references 4 -7). Also, reference 146 is not used correctly, since the author stated that the success of these interventions in pre-clinical and clinical studies has brought huge hope in improving female and male reproductive health and the title of the referring article is Research progress on the treatment of premature ovarian failure using MSCs. Neither references 155 nor 156 include results from MSC transplantation. Do the authors think that spermatogonial stem cells are MSCs?
Table summarizing MSCs isolated from different tissues used in animal models used for infertility treatment includes only bone marrow-derived and adipose tissue-derived MSCs and does not cover MSCs isolated from other sources, such as menstrual blood, endometrial tissue, umbilical cord, umbilical cord blood, amniotic fluid, placenta – which are relevant for infertility treatment. On the contrary, spermatogonial and oogonial stem cells are included – but these are not MSCs!
Mitochondria transfer is not mentioned in the section summarizing MSC properties. And immunomodulatory and anti-inflammatory capacity of MSCs is related only to exosome production (line 289).
In section 6, the use of organoids is discussed, unfortunately not in the context of MSCs.
Extensive editing of English language and style required, some sentences have no sense (for example Since the concept of pluripotent stem cells could differentiate into functional gametes, few primordial germ cells have been associated with the recruitment and stimulation or conversion into functional gametes – lines 353-4).
I agree with the authors that stem cells provide an exciting opportunity in infertility therapies, however, the manuscript suffers from many problems and needs extensive revision before it can be published in the journal Biology. Mainly include all references and use them correctly! Focusing only on MSCs, shortening the part introducing diseases leading to infertility and covering MSCs originating from tissues other than bone marrow and adipose tissue.
Author Response
Response to Reviewer 2 Comments
Point 1: The authors presented a manuscript summarizing current knowledge on the use of mesenchymal stem cells (MSCs) in female and male infertility therapies. Overall, this topic is important, however, several articles have been published recently that review the current knowledge on this topic. For example? Fazeli et al. Mesenchymal Stem Cells (MSCs) Therapy for Recovery of Fertility: a Systematic Review DOI10.1007/s12015-017-9765-x. Saeed et al Mesenchymal stem cells to treat female infertility; future perspective and challenges: A review DOI10.18502/ijrm.v20i9.12061; Saha et al. Application of Stem Cell Therapy for Infertility DOI10.3390/cells10071613; Jahanabani et al. Stem cells technology as a platform for generating reproductive system organoids and treatment of infertility-related diseases DOI10.1002/cbin.11747. (These articles are not mentioned in the presented manuscript!) Therefore, it is very difficult to find a specific field that has not been covered by recent reviews and in my opinion, the authors failed in this respect.
Thank you for taking your precious time in reviewing our manuscript. The above suggested review papers have been incorporated in our manuscript accordingly in line 59-61. However, we would like to emphasise that our study focuses explicitly on the autologous MSCs treatment in infertility, which is lacking in the other reviews. Most review studies focuses largely either in one of the themes; infertility and stem cells, gender-based infertility and stem cells or challenges and future perspective of infertility with stem cells. In terms of autologous MSC-based therapy infertility and the future perspective has not yet been covered in the literature, thus the reason we produce a review focuses on this field. We hope that the readers could gain more inputs on autologous stem cells treatments from our review and fills the gap of this lacking review in the literature since most studies focus both on autologous and allogeneic treatments.
Point 2: The introduction of diseases leading to infertility is too long and is not associated with the topic – using of MSCs.
We have shortened Section 2: Causes and Mechanisms Leading to Infertility. We only highlighted the main causes that could lead to infertility and retain the table summarising the causes of infertility.
Point 3: The necessity of stem cell application in the treatment of both female and male infertility is supported by the articles focusing only on male infertility (references 4 -7).
Thank you for pointing the lack of reference/s in this statement. We have added a reference from one of the references that you suggested above, Saeed paper on the review of mesenchymal stem cells in female infertility.
Point 4: Also, reference 146 is not used correctly, since the author stated that the success of these interventions in pre-clinical and clinical studies has brought huge hope in improving female and male reproductive health and the title of the referring article is Research progress on the treatment of premature ovarian failure using MSCs.
Thank you for the comment. We have amended the reference by adding one of the reference that you suggested earlier; Fazeli et al on the MSCs recovery of infertility.
Point 5: Neither references 155 nor 156 include results from MSC transplantation.
Thank you for your comment. Reference 155 has now has changed to 134. After rechecking, the paper did transplantation work on pre-pubertal monkeys in the report, in which they showed differed testicular growth between non-transplanted and transplanted testes (under the headings of “Testicular size and sperm samples after irradiation”). Therefore, I have retained reference 134 and have removed the reference 155.
Point 6: Do the authors think that spermatogonial stem cells are MSCs?
Thank you for your comment. After a discussion among the authors, we have decided to remove spermatogonial stem cells (SSCs), oogonial stem cells (OSCs) and thecal stem cells from the manuscript. We have only retained BMSCs and ADSCs in discussing the focus point of our manuscript, which is the autologous MSCs-based therapy in infertility. Therefore, SSCs, OSCs and thecal stem cells have been removed from Table 2 and Table 3. We touched a little bit on the germline stem cells (SSCs and OSCs) in the Subheading 5.1 as one of autologous stem cells treatment that can help in infertility.
Point 7: Table summarizing MSCs isolated from different tissues used in animal models used for infertility treatment includes only bone marrow-derived and adipose tissue-derived MSCs and does not cover MSCs isolated from other sources, such as menstrual blood, endometrial tissue, umbilical cord, umbilical cord blood, amniotic fluid, placenta – which are relevant for infertility treatment. On the contrary, spermatogonial and oogonial stem cells are included – but these are not MSCs!
Thank you for the comment. We did not cover MSCs from other sources since our aim in this manuscript is reviewing the use of AUTOLOGOUS MSCs. Therefore, it is impossible to use umbilical cord, umbilical cord blood, amniotic fluid and placenta MSCs in the concept of autologous MSCs treatment for infertility. We have added endometrial tissue stem cells in the manuscript. And as for the spermatogonial and oogonial stem cells, after a careful consideration from all of the authors, we have removed it from the manuscript.
Point 8: Mitochondria transfer is not mentioned in the section summarizing MSC properties. And immunomodulatory and anti-inflammatory capacity of MSCs is related only to exosome production (line 289).
Thank you for the comment. We have added the mitochondria transfer in the summary of MSCs properties. While mitochondria transfer acted via cell-to-cell interaction, immunomodulatory and anti-inflammatory properties of MSCs acted via their secretomes mainly exosomes and inflammatory mediators covered in the manuscript.
Point 9: In section 6, the use of organoids is discussed, unfortunately not in the context of MSCs.
Thank you for the comment. After reviewing the document, we have decided to remove the organoids section from our manuscript. Section 6.3 is now has been changed to “Regenerative therapy”.
Point 10: Extensive editing of English language and style required, some sentences have no sense (for example Since the concept of pluripotent stem cells could differentiate into functional gametes, few primordial germ cells have been associated with the recruitment and stimulation or conversion into functional gametes – lines 353-4).
Thank you for your comment. We have made the sentence simpler and easier to read and understand. We have also carefully read the manuscript for any grammatical mistakes, which is done by all of the authors and a second set of eye who is a native English-speaking colleague.
Point 11: I agree with the authors that stem cells provide an exciting opportunity in infertility therapies, however, the manuscript suffers from many problems and needs extensive revision before it can be published in the journal Biology. Mainly include all references and use them correctly! Focusing only on MSCs, shortening the part introducing diseases leading to infertility and covering MSCs originating from tissues other than bone marrow and adipose tissue.
Thank you for your comment. We have included the suggested references appropriately in our manuscript (where it fits) and we have revised the references that have been pointed out to be incorrectly used in the manuscript. However, there is one reference that we have rebutted in our point no.5. We have also shortened the introduction of diseases leading to infertility. However, we cannot include MSCs other than bone marrow or adipose tissues since our manuscript mainly focuses on the AUTOLOGOUS MSCs and this technique is impossible to be done on embryonic stem cells, umbilical cord stem cells and other stem cells.
Reviewer 3 Report
The review demonstrates the current problems in treating human infertility, especially for the cases with “unexplained causes” and proposes that the usage of stem cells, mesenchymal stem cells in particular, is one of the promising future directions. The review is very well organized and presents the current status and data in various causes of human infertility and on-going basic and clinical research in relatively clear fashion. There are, however, several issues could be addressed:
1) The concept of MSCs and other types of tissue stem cells are sometimes mixed and inter-converted in the text, which may cause confusions, for example, lines 249-251, references cited are not precisely on PGCs but on spermatogonial stem cells; lines 357-358, treated MSCs the same as spermatogonial stem cells. These should be checked and corrected throughout the text.
2) While Section 5 and 6 present the current knowledge of on-going research in both animal models and clinical settings, a lack of summary or analysis of the potential targets of MSC application makes it hard to follow as why need MSC-based therapy. It may be helpful if the authors could provide a brief summary at the beginning of Section 5 to illustrate the goal of using stem cell technology in treating human infertility, whether for its potential to derive gametes or for its usage in generating auxiliary factors to improve the functional roles of various cell types in the reproductive systems.
3) Some other minor factors: can the title of Figure 1 be changed to relevance of infertility causes and MSC usage, rather than referring to the review; the Figure 2 may be enlarged for better readability; some explanation of “introduction of heterogeneity” in line 426.
Author Response
Response to Reviewer 3 Comments
Point 1: The concept of MSCs and other types of tissue stem cells are sometimes mixed and inter-converted in the text, which may cause confusions, for example, lines 249-251, references cited are not precisely on PGCs but on spermatogonial stem cells; lines 357-358, treated MSCs the same as spermatogonial stem cells. These should be checked and corrected throughout the text.
Thank you for your comment. We have changed the PGCs term in line 186-188 (previously 249-251) to “germline stem cells” that fits perfectly with the term of spermatogonial stem cells and ovarian stem cells. In line 307-316 (previously 357-358), we have changed the PGCs term to germline stem cells. In the context of treating spermatogonial stem cells as MSCs, all authors have agreed (and also upon suggestion from reviewer 1) to delete the spermatogonial stem cells and oogonial stem cells from the table. But we decided to remain the example of the success of this stem cells in the text showing that there are other mechanisms of autologous treatment that can be done to treat infertility but is not categorised as MSCs.
Point 2: While Section 5 and 6 present the current knowledge of on-going research in both animal models and clinical settings, a lack of summary or analysis of the potential targets of MSC application makes it hard to follow as why need MSC-based therapy. It may be helpful if the authors could provide a brief summary at the beginning of Section 5 to illustrate the goal of using stem cell technology in treating human infertility, whether for its potential to derive gametes or for its usage in generating auxiliary factors to improve the functional roles of various cell types in the reproductive systems.
Thank you for your suggestion. We have added a summary sentence in the paragraph. Line 276-281:
“Exhaustion of traditional treatments and genetic defects that lead to gamete deficiency have led to mounting preclinical studies on animals and clinical studies on humans using stem cells one of which is using MSCs. This MSCs therapy-based technology could generate auxiliary factors in improving the functional roles of various cell types in the reproductive systems”.
Point 3: Some other minor factors: can the title of Figure 1 be changed to relevance of infertility causes and MSC usage, rather than referring to the review; the Figure 2 may be enlarged for better readability; some explanation of “introduction of heterogeneity” in line 426.
Upon the auditor comment, we have decided to change Figure 1 into Graphical Abstract. Figure 2 image has been enlarged in the document. And we have added some explanation to the “heterogeneity” term in line 387-389 (previously 426).
Round 2
Reviewer 2 Report
The authors presented a revised version of the manuscript summarizing current knowledge on the use of mesenchymal stem cells (MSCs) in female and male infertility therapies. The authors explained that their manuscript differs from several publications covering this area by aiming to the role of autologous MSCs. Therefore, this fact should be reflected in the manuscript title. Also, the Introduction lacks an explanation of why the use of autologous MSCs is more advantageous, or whether they have any disadvantages compared to allogeneic MSCs, although it is explained on page 8.
Chapter 3 – Limitations in the current conception treatment are important, but still too long. A detailed explanation of the individual problems of current infertility treatment is not important for the topic of the manuscript.
I do not agree with the authors that autologous MSCs derived from umbilical cord blood could not be used in infertility treatment. There was a huge campaign in Europe to collect and store umbilical cord blood for possible treatment in the future.
Several publications showed mitochondrial transfer as an important mechanism of MSC function (reviewed for example here doi: 10.1042/BSR20182417). Furthermore, Sertoli cells have been shown to possess a quality of MSCs (PMCID: PMC5375999.), including the ability to transfer mitochondria (DOI: 10.1007/s12015-021-10197-9).
Overall, the quality of the revised manuscript improved. I propose further shortening of Chapter 3. If the authors focused on autologous MSCs, this should be reflected in the title, Introduction, and also in Conclusion.
Author Response
Dear Reviewer,
Thank you for your time in reviewing our re-submitted manuscript. We very much appreciate your positive feedbacks and suggestions in improving our manuscript. Below are our responses in view of your recent comments in regards to our manuscript;
Comment 1:
The authors presented a revised version of the manuscript summarizing current knowledge on the use of mesenchymal stem cells (MSCs) in female and male infertility therapies. The authors explained that their manuscript differs from several publications covering this area by aiming to the role of autologous MSCs. Therefore, this fact should be reflected in the manuscript title.
Thank you for your suggestion. We have changed our title to “Autologous Human Mesenchymal Stem Cells-Based Therapy in Infertility: New Strategies and Future Perspectives”.
Comment 2:
Introduction - Also, the Introduction lacks an explanation of why the use of autologous MSCs is more advantageous, or whether they have any disadvantages compared to allogeneic MSCs, although it is explained on page 8.
Thank you for highlighting this point. We have now added a brief explanation in the introduction section where you can find it in line 61-64.
Comment 3:
Chapter 3 – Limitations in the current conception treatment are important, but still too long. A detailed explanation of the individual problems of current infertility treatment is not important for the topic of the manuscript.
We have taken into a consideration on your comment. But after a thorough discussion with all authors, we have decided to shortened Chapter 3 (by eliminating some sentences) but to retain the examples of issues in the limitation of the current conception treatment. The authors think that we need to link the reader of these limitations with the alternative cell-based therapy of MSCs that are available and to give a better picture of autologous treatment of MSCs.
Comment 4:
I do not agree with the authors that autologous MSCs derived from umbilical cord blood could not be used in infertility treatment. There was a huge campaign in Europe to collect and store umbilical cord blood for possible treatment in the future.
Thank you for your comment. In the manuscript, line 319 (page 7):
“Manipulation of BMSCs, Umbilical Cord Stem Cells, Amniotic Fluid Mesenchymal Stem Cells, Menstrual Stem Cells (MenSCs), Adipose-Derived Stem Cells (ADSCs), and endometrial MSCs with the options using autologous or allogeneic treatments proved the effectiveness of MSCs in infertility”
we did mention that umbilical cord stem cells as one of the options in treating infertility and we did not entirely eliminate the possibility of autologous treatment of umbilical cord stem cells. However, in this paper, thus far, we have only been focusing on the MSC treatment that have been performed either in animal models or in human with published studies. Therefore, the authors agree that any future treatment that has not been proved or published is not included in this paper.
Comment 5:
Several publications showed mitochondrial transfer as an important mechanism of MSC function (reviewed for example here doi: 10.1042/BSR20182417). Furthermore, Sertoli cells have been shown to possess a quality of MSCs (PMCID: PMC5375999.), including the ability to transfer mitochondria (DOI: 10.1007/s12015-021-10197-9).
Thank you for your input on Sertoli cells. We have incorporated the suggested papers in our manuscript, which you can find it in line 277-281.
Comment 6:
Overall, the quality of the revised manuscript improved. I propose further shortening of Chapter 3. If the authors focused on autologous MSCs, this should be reflected in the title, Introduction, and also in Conclusion.
Thank you very much for your positive feedback. We have taken our action in shortening Chapter 3 (point no. 3) and have included the word autologous in our title (point no.1), a brief explanation why we chose autologous in the introduction (point no.2), and a statement in the conclusion.